# Study of the Self-Healing Performance of Semi-Flexible Pavement Materials Grouted with Engineered Cementitious Composites Mortar based on a Non-Standard Test

**DOI:** 10.3390/ma12213488

**Published:** 2019-10-24

**Authors:** Xu Cai, Wenke Huang, Kuanghuai Wu

**Affiliations:** School of Civil Engineering, Guangzhou University, Guangzhou 510006, China; cx_caixu@163.com (X.C.); hwk_gzu@163.com (W.H.)

**Keywords:** self-healing pavement materials, semi-flexible pavement, ECC mortar, crack resistance, curing conditions

## Abstract

Semi-flexible pavement (SFP) materials, with their characteristics of good high temperature stability, strong durability, and lower cost, are suitable for heavy-duty roads, but their cracking problem has hindered the development and popularization of this kind of pavement to a certain extent. In this study, engineered cementitious composites (ECC) were used to form ECC-SFP materials. The self-healing properties of ECC-SFP materials with three kinds of voids of matrix asphalt mixtures were studied. The test results showed that the fluidity and strength of the ECC mortars met the specification requirements when the water–cement ratio was 0.23 and the ECC fiber dosage was 1–2%. The flexural strength of ECC mortar is better than that of ordinary mortar. The higher the ECC fiber dosage, the higher the flexural strength. Increasing the void of the matrix asphalt mixture and the amount of ECC mortar increased the toughness of the ECC-SFP material, which was seen as an increase of the flow value. Curing conditions are key factor affecting the self-healing properties of ECC mortar and ECC-SFP materials. The self-healing effect of materials in 60 °C water is the best. When an ECC fiber dosage of 1% was used, the *HI*_mor_ of ECC mortar and *HI*_mix_ of ECC-SFP material were 27.5% and 24.8%, respectively. With the addition of ECC material, ECC-SFP material achieved a certain degree of self-healing, but this still needs to be further optimized. Studies of grouting process optimization and increasing the ECC fiber dosage are feasible directions to explore in order to improve the self-healing properties of ECC-SFP materials in the future.

## 1. Introduction

Semi-flexible pavement (SFP) material is a kind of pavement material which is composed of a special cement mortar infused in the large voids of an open-graded matrix asphalt mixture [1,2]. Its high temperature stability is better than that of asphalt pavement, and it has superior deformation resistance, water damage resistance, and skid resistance. Therefore, SFP material is mainly used in long steep slope sections, tunnels, small radius curve roads, toll crossings, urban road intersections, and at bus stops [3,4,5,6,7].

However, due to the presence of asphalt, cement mortar and aggregate, SFP materials contain interface areas composed of different materials. The differences in the physical and mechanical properties of these materials makes it easy to produce stress concentration inside the SFP materials, which makes them prone to crack damage. At present, research on the cracking of SFP materials usually focuses on the properties of asphalt, aggregate, and cement mortar, and their influences on cracking resistance. For example, Husain [8] studied a cementing material’s aggregate grading, durability and strength through statistical analysis, and obtained the influence of gradations in the performance of SFP materials. Pei [9] added different kinds and amounts of additives into the mortar, and found that water reducer, expansion agents, and air-entraining agents had different effects on fluidity, strength, and shrinkage. Wang [10] tested nine mixtures to find the relationship between compressive strength, rupture strength, the water–cement ratio, and the sand-cement ratio. The test indicted that both the cement mortar and the asphalt skeleton affected the performance of the SFP materials. For the cement mortar, the water–cement ratio and sand-cement ratio had great influences on its performance, such as the workability, mechanical strength, and volume stability, and therefore in the performance of the SFP materials. Ling [5] applied SFP materials with asphalt-rubber to a heavy-duty test road, and both laboratory tests and the test road showed that the SFP with asphalt-rubber achieved an excellent performance. Suhana [11] investigated the mechanical properties of cement-bitumen composites as an alternative SFP surfacing material. The findings showed that by replacing 5% of the cement with silica fume there was an improvement in compressive strength and the tensile stiffness modulus. Yang [12] conducted a runway test bench by using eight groups of SFP specimens. The results showed that the maternal asphalt mixture void content was the most important factor that affected durability to cyclic wheel load. When the maternal asphalt mixture void content was 26%, the mechanical performance of the semi-flexible material was superior. Zhang [13] studied the effects of composition and formulation on grouting material. The results showed that cement paste, with its optimal formulation, was more suitable as a grouting material and achieved better performance. The optimal ratio of water to cement was 0.58, content of coal ash was 10%, and content of mineral powder was 10%. Hong [14] evaluated the freeze–thaw durability of SFP mixes grouted with early-strength (ES) and high-strength (HS) grouting materials. The results showed that enhancement of the freeze-thaw resistance of an SFP mix can be achieved by incorporating a high-strength grouting material.

Although most studies have revealed the influencing factors of cracking of SFP materials, it is still difficult to effectively avoid cracking. In fact, the asphalt component in SFP material has certain self-healing properties, meeting the basic conditions of self-healing design. Recently, research into pavement self-healing has been paid more attention. Sun [15] pointed out that self-healing capability for repairing micro-crack damage can restore functionality, at least to some extent, and considerably reduce maintenance costs, as well as extend the pavement’s service life and eventually decrease the emissions of greenhouse gas from pavement production. The self-healing of asphalt materials can be described as the partial restoration of the intrinsic asphalt structure across adjacent crack surfaces [16]. Mohammad [17] evaluated the effects of a combination of nano-silica and styrene-butadiene-styrene (SBS) on the self-healing ability of hot mix asphalt (HMA) by applying an indirect tension test (IDT). The main reason for crack healing was the flow of bitumen mortar into micro-cracks under gravity forces. Giorgia [18] evaluated the self-healing potential and thixotropy of bituminous mastics. The results showed that the self-healing properties of the material were not significantly related to the degree of aging of the asphalt. An aged bitumen content of up to 45% (45R) improved the overall fatigue response without hindering self-healing capability. Xu [19] explored the potential use of calcium alginate capsules in porous asphalt. The results showed that samples with capsules were able to achieve a healing index that was 6% higher than the reference samples. Wang [20] estimated the fatigue and healing characteristics of asphalt binder by newly developed linear amplitude sweep (LAS) and LAS-Based Healing (LASH) protocols. The combined use of LAS and LASH tests is recommended for effectively distinguishing and designing the fatigue-healing performance of neat and modified asphalt binders. Fan [21] evaluated the fracture resistance and self-healing performance of asphalt concrete at low temperatures using a semicircular bending (SCB) test. The results showed that the optimum healing temperature was 60 degrees at a healing time of 8 h.

To broadly define the healing phenomenon, healing acts as long as the material does not completely fail and there is still contact between the crack faces, since an external force can be applied to make two fractured surfaces come into contact [16,22]. Due to the existence of asphalt mortar, the cracking of SFP materials is less than that of cement pavement, and the crack width is generally smaller [2,23]. Therefore, SFP material has good self-healing design potential. Since the matrix asphalt mixture has a good self-healing property, the emphasis of the self-healing design of SFP material is to control the crack width and the self-healing design of cement paste.

In fact, each failure and self-healing recombination is the concentration and dissipation of internal stress in the material. After repeated failure and healing cycles, the rigid and flexible components in SFP materials will have better collaborative deformation capacity. Based on this concept, this study used engineered cementitious composites (ECC) mortar to replace the original cement mortar. The mixing ratio of ECC mortar was determined through experiments, and the self-healing properties of ECC mortar and SFP materials grouted with ECC mortar under different curing conditions were studied.

## 2. Experimental Design

According to the predetermined research objectives, we developed the following experimental program.

### 2.1. Materials

Materials used in this study include asphalt, aggregates, powder mortar and engineered cementitious composites.

#### 2.1.1. Asphalt and Aggregates

Shell 70# asphalt was selected for this test. The technical indexes of asphalt are shown in Table 1. The aggregates were formed from granite from the Guangdong Furong quarry (Shenzhen, China), and the technical indicators are shown in Table 2.

#### 2.1.2. Powder Mortar

The powder mortar used in this study is based on the research results obtained by our research team [24,25,26], and is composed of cement, fine sand, fly ash, mineral powder, poly-carboxylic acid water-reducing agent, and other additives. The composition of the mortar is shown in Table 3.

#### 2.1.3. Engineered Cementitious Composites

The performance of fiber in a cement mortar mainly depends on its strength, elongation, elastic modulus, and other mechanical properties [27,28,29]. Considering the economic technology and feasibility of the test, PVA-ECC fiber was used in the preparation of the ECC mortar in this paper. The physical parameters of the ECC are shown in Table 4.

### 2.2. Matrix Asphalt Mixture

The matrix asphalt mixture not only needs to have the required strength, but also have enough connected and semi-connected voids to ensure the grouting of the cement mortar when forming a semi-flexible structure. In this study, the volume method was used to design three matrix asphalt mixtures with voids of 20%, 25%, and 28%. The asphalt mixtures were expressed by MA20, MA25, and MA28, respectively. The optimal asphalt contents were determined by the Cantabro Test and Shellenburg leakage test. The design results are shown in Table 5.

### 2.3. Engineered Cementitious Composites(ECC) Mortar

Cement mortar used for SFP materials should have better fluidity, higher strength, and greater volume stability. According to the existing research results [30,31,32], this study used a cement mortar mixer to prepare the ECC mortar, as shown in Figure 1. The ECC was soaked in water before mixing to ensure that the ECC fully absorbed the water. There were four steps to prepare the ECC mortar: (1) the dry powder mortar was blended at a low speed for 1 min; (2) 2/3 water was added and blended at high speed for 4 min; (3) another 1/3 water was added and blended at high speed for 4 min; (4) the ECC was added and blended for 2 min to disperse the fiber evenly in the mortar.

#### 2.3.1. Test of Fluidity

The fluidity was tested by a special v-shaped funnel. The upper diameter of the funnel was 178 mm, the lower diameter was 13 mm, the outflow pipe length was 38 mm, and the effective volume was 1000 mL, as shown in Figure 2. When the discharge time of 1000 mL mortar from the funnel was 12–18 s, it was considered that the mortar met the requirement for fluidity [24]. Mortars with a water–cement ratio of 0.20, 0.21, 0.22, 0.23 and 0.24 were tested, and 0%, 1%, 2% and 3% ECC fibers were added into the mortar. Three parallel experiments were conducted for each proportion, and the average value was used as the final test result.

#### 2.3.2. Strength Test for Mortar

According to the fluidity test results, the ECC mortar ratio with the desired fluidity performance was selected, and the flexural and compressive performance test [33] was carried out according to the specifications. Specimens of 40 mm × 40 mm × 160 mm were used in the flexural strength test, as shown in Figure 3. Under the same water–cement ratio, molding mode and curing environment, the flexural strength of the ECC mortar and ordinary mortar with different fiber contents of 3 days, 7 days, 14 days and 28 days was measured.

According to the Test Methods of Cement and Concrete for Highway Engineering, the specimen used in the compression test was the specimen selected after the flexural test, and the compression surface was the side of the specimen during molding. The loading rate of the press was set at 2400 N/s ± 200 N/s until the specimen was damaged. The press and test fixtures used are shown in Figure 4.

#### 2.3.3. Self-Healing Test of Mortar

According to the strength test results, the optimal mix ratio of ECC mortar was selected. The self-healing properties of the ECC mortar were evaluated based on the shear strength test. Firstly, specimens of 40 mm × 40 mm × 160 mm were prepared and maintained by conventional means for seven days. Secondly, according to the shear strength test method, the sample was preloaded at the rate of 1 mm/min. The load was controlled to reach 50% of the material’s 7 day flexural strength, and the preloading specimen was obtained, as shown in Figure 5.

Three different curing conditions were adopted for the specimens with cracks, and the curing time was 21 days. The first condition was dry air curing. The specimen was placed in a dry environment with a humidity of 10% at 20 °C. The second condition was wet air curing, in which the specimen was placed in an environment of 20 °C with 100% humidity. The last was hot and humid curing. The specimen was placed in a 60 °C constant temperature water tank. After curing, the flexural strength of the specimen was measured.

During the test we found the strength of cement mortar varies from 30% to 50% between 7 days and 28 days. Considering that the strength of cement mortar will increase with the increase of curing time, and the self-healing ratio calculated using the ratio of the strength of preload specimens at 28 days and 7 days cannot eliminate the contribution of cement hydration. Therefore, the self-healing property of the material was characterized by the ratio of the 28 days flexural strength of the preloaded specimen to the 28 days flexural strength of a standard specimen in this study, as shown in Equation (1).
(1)$$HI_{\mathrm{mor}}=\frac{{R_{f}}^{\prime }}{R_{f}}$$
where $HI_{\mathrm{mor}}$ is the self-healing ratio of flexural strength; ${R_{f}}^{\prime }$ is the 28 days flexural strength of the preloaded specimen; and $R_{f}$ is the 28 days flexural strength of the standard specimen.

### 2.4. The Semi-Flexible Pavement (SFP) Materials

According to the test results of the ECC mortar, the optimal mix ratio of the ECC mortar was selected to carry out the self-healing performance test of the SFP material, including the Marshall stability test and asphalt mixture three-point bending test.

#### 2.4.1. Specimen Grouting

The matrix asphalt mixture specimen was first formed and the bottom and sides of the specimen were sealed with transparent tape. Then, mortar was poured onto the surface of the specimen, and a vibration table was used to assist the grouting. The vibration time was generally controlled at about 90 s. During grouting, the surface of the specimen was observed until there was no bubbling. Excess grout was scraped off the surface after grouting. The grouting effect is shown in Figure 6.

#### 2.4.2. Marshall Stability Test

A batch of matrix asphalt mixture was formed according to the designed voids of 20%, 25%, and 28%. The ECC mortar and ordinary mortar were respectively grouted, and the specimens were cured for 3 days, 7 days and 28 days, respectively. Four Marshall specimens of each mixing ratio were tested under the condition of 60 °C. The test results are expressed as averages.

#### 2.4.3. Three-Point Bending Test

According to the Standard Test Methods of Bitumen and Mixtures for Highway Engineering [34], the 300 mm × 300 mm × 60 mm rutting specimens of matrix asphalt mixtures with voids of 20%, 25%, 28% were formed, and the ECC mortar and ordinary mortar were respectively grouted. After 7 days of curing time, the rutting specimens were cut into 250 mm × 30 mm × 35 mm beam specimens, as shown in Figure 7. The beam specimens were divided into two groups. One group was maintained for 7 days, and the other group was maintained for 28 days. The three-point bending test was carried out after curing. 

In the three-point bending test, the beam specimen was placed on a bracket with a span of 200 mm, and vertical load was applied at the central point of the beam. The test temperature was –10 °C and the loading rate was 50 mm/min. The calculation of the three-point bending strength, $R_{B}$, the maximum bending strain, $\varepsilon_{B}$, and the stiffness modulus, $S_{B}$, are shown in Equations (2)–(4).
(2)$$R_{B}=\frac{3\;\times \;L\;\times \;P_{B}}{2\;\times \;b\;\times \;h^{2}}$$
(3)$$\varepsilon_{B}=\frac{6\;\times \;h\;\times \;d}{L^{2}}$$
(4)$$S_{B}=\frac{R_{B}}{\varepsilon_{B}}$$
where $R_{B}$ is the three point bending strength of the specimen; $\varepsilon_{B}$ is the maximum bending strain; $S_{B}$ is the stiffness modulus of the materials; *b* is the width of the specimen; h is the height of the specimen; L is the span of the specimen; $P_{B}$ is the maximum load when the specimen fails; and *d* is the mid-span deflection of the specimen when it fails.

#### 2.4.4. Self-Healing Test of the Semi-Flexible Pavement (SFP) Materials

The self-healing properties of the SFP materials were evaluated based on the three-point bending test. Beam specimens were formed according to the method outlined in Section 2.4.3, and the specimens were preloaded at the rate of 1 mm/min under the condition of –10 °C for 7 days of curing time. The load was controlled to reach 50% of the material’s 7 days three-point bending strength, and preloaded specimens were obtained. After that, the three curing conditions mentioned in Section 2.3.3 were used for curing for 21 days. After curing, the three-point bending strength of the SFP beam specimens at 10 °C was measured. Based on the same considerations as the calculation of self-healing ratio of mortar, the self-healing properties of the SFP materials were characterized by the ratio of the 28 days three-point bending strength of the preloaded specimens to the 28 days three-point bending strength of standard specimens, as shown in Equation (5).
(5)$$HI_{\mathrm{mix}}=\frac{{R_{B}}^{\prime }}{R_{B}}$$
where $HI_{\mathrm{mix}}$ is the self-healing ratio of the SFP materials; ${R_{B}}^{\prime }$ is the 28 days three-point bending strength of the preloaded SFP beam specimens; and $R_{B}$ is the 28 days three-point bending strength with the standard test.

## 3. Results and Discussion

According to the proposed tests, the following results can be obtained.

### 3.1. Mortar

#### 3.1.1. Fluidity

According to the designed fluidity test, mortars with water–cement ratios of 0.20, 0.21, 0.22, 0.23, and 0.24 were tested separately, and ECC fiber dosages of 1%, 2%, and 3% was added to the mortar for the fluidity test. The results are shown in Table 6.

According to Table 6, the fluidity of the mortar deteriorates after the addition of the ECC, and the flow time increases. Under the same water–cement ratio, the fluidity decreases with the increase of ECC content. Secondly, the fluidity increases as the water–cement ratio increases. When the ECC content reaches 3%, there is too much flocculent in the mortar, and the water–cement ratio needs to reach 0.24 to meet the minimum requirement of fluidity. When 1% and 2% ECC are mixed in, the fluidity value of the mortar basically meets the requirements of 12–18 s flow time. The flow state of the mortar is shown in Figure 8.

According to the fluidity test results, the dosage of ECC should not be more than 2%. At the same time, in order to increase the fluidity of the ECC mortar, a larger water–cement ratio should be used. The fluidity of 2% ECC mortar is less than 12 s when the water–cement ratio is 0.24, which does not meet the requirements of the specification. Therefore, a water–cement ratio of 0.23 was adopted in subsequent studies to carry out mechanical tests on mortars with ECC fiber dosages of 0%, 1%, and 2%.

#### 3.1.2. Flexural Strength and Compressive Strength of Mortar

According to ECC fiber dosages of 0%, 1% and 2%, four groups of 40 mm × 40 mm × 160 mm test pieces were formed. There were three specimens in each group, for a total of 36 specimens. Four groups of specimens with the same ECC fiber dosage were cured for 3 days, 7 days, 14 days, and 28 days. The flexural strength and compressive strength were tested after curing. The test results are shown in Table 7.

According to Table 7, under the same ECC fiber dosage, the flexural strength and compressive strength of the cement mortar increased with the increase of curing time. The flexural strength and compressive strength after curing for 7 days were 65% and 70% of that found after curing for 28 days, respectively. Secondly, with the increase of ECC fiber dosage, the flexural strength of the cement mortar increased but the compressive strength decreased. The 28 days flexural strengths of 1% and 2% ECC mortar were 12% and 16% higher than that of mortar without ECC, and the compressive strengths were 98% and 97% of the mortar without adding ECC. The addition of ECC caused the specimen show a certain degree of toughness and increased the crack resistance of the material. Failure modes of the specimens are shown in Figure 9.

Figure 9 indicates that the ECC performed as a bridge-connection effect when the specimen was damaged, so that the specimen could still be connected through the ECC after becoming cracked. The ECC mortar exhibited stronger tensile resistance, higher toughness, and greater energy absorption capacity than ordinary mortar. 

#### 3.1.3. Self-Healing Evaluation of Mortar

Nine specimens of 40 mm × 40 mm × 160 mm were prepared with the ECC fiber dosages of 0%, 1% and 2%. The mortar self-healing test was carried out according to the method mentioned in Section 2.3.3, and the test results are shown in Table 8. 

The mortar without ECC were completely fractured at the preloading stage, so further test data could not be obtained. As can be seen from Table 8, curing conditions and ECC content were the main factors affecting the self-healing performance of ECC mortar. The recovery of specimens under the three curing conditions varied. The $HI_{\mathrm{mor}}$ for curing in dry air was 5.5–6.5%. The $HI_{\mathrm{mor}}$ for curing in moist air was 13.5–16.0%. However, in a hot and humid environment, the maximum value of $HI_{\mathrm{mor}}$ was obtained, of 27.5–30.0%. The increase of ECC fiber dosage slightly improved the self-healing property of ECC mortar. Under the same curing conditions, the self-healing performance of ECC mortar with 2% dosage was slightly better than that of ECC mortar with 1% dosage. 

In conclusion, the self-healing performance of ECC mortar is related to environmental temperature, environmental humidity, and ECC fiber dosage. At 20 °C, increasing ambient humidity increases $HI_{\mathrm{mor}}$ by about 8.5%. Raising the ambient temperature to 60 ℃ in wet conditions increases the $HI_{\mathrm{mor}}$ by about 14%. Increasing the dosage of ECC can increase $HI_{\mathrm{mor}}$ by 1.3–3.3%. Therefore, the curing condition is the main factor affecting the self-healing effect of ECC mortar. The recommended dosage of ECC is 1%.

### 3.2. Semi-Flexible Pavement (SFP) Materials

According to the self-healing test results of ECC mortar, 0% and 1% ECC fiber dosage were selected to prepare the cement mortar. Marshall specimens and three-point bending specimens of SFP materials were prepared with matrix asphalt mixture with voids of 20%, 25%, and 28%.

#### 3.2.1. Marshall Test

Twenty-four Marshall specimens of matrix asphalt mixture for each void ratio were filled with cement mortar with ECC fiber dosages of 0% and 1%, respectively (called SFP material and ECC-SFP material, respectively) to test the Marshall stability and flow value. The test results are shown in Figure 10.

According to Figure 10a, the void ratio of matrix asphalt mixture was the main factor affecting the stability of the material. The Marshall stability increased with the increase of the void ratio of the matrix asphalt mixture. Secondly, the Marshall stability increased with the increase of curing time, and the Marshall stability at 3 and 7 days reached 45% and 70% respectively, as it was at 28 days. However, ECC has little effect on the Marshall stability. Under the same ratio of air voids and curing time, the difference in the Marshall stability between ECC-SFP materials and SFP materials was not obvious.

According to Figure 10b, the flow value decreased gradually with the increase of curing time. The void ratio of the matrix asphalt mixture also affected the trend of the flow value. The flow value of ECC-SFP materials increased with the increase of the void ratio, while that of SFP materials decreased with the increase of the void ratio. Meanwhile, the flow value of the ECC-SFP material was higher than that of the SFP material, which indicated that ECC can increase the toughness of SFP materials and give them a better deformation capacity.

#### 3.2.2. Three-Point Bending Test

Four rutting specimens were molded for each void ratio of matrix asphalt mixture, and the ECC mortar and ordinary mortar were grouted. After 7 days of curing, the rutting specimens were cut to obtain beam specimens. The three-point bending test was carried out according to the method in Section 2.4.3. The results of the three-point bending test at 7 days and 28 days are shown in Figure 11.

According to Figure 11a, the three-point bending strength increased with the increase of the void ratio of the matrix asphalt mixture. Secondly, the use of ECC mortar can improve the three-point bending strength. Under the condition of 28 days of curing time, the bending modulus values of the ECC-SFP materials with three voids were 7.2%, 23.1% and 17.8% higher than for the SFP materials, respectively.

According to Figure 11b, the void ratio of the matrix asphalt mixture affected the maximum bending strain. The maximum bending strain of the ECC-SFP material increased with the increase of the void ratio, reaching the maximum value at 25% void. Meanwhile, the maximum bending strain of SFP materials decreased with the increase of the void ratio. Secondly, the grouted amount of ECC mortar significantly increased the maximum bending. The larger the void ratio of the matrix asphalt mixture, the more ECC mortar was grouted, and the larger the maximum bending strain. Under the condition of 28 days of curing time, the maximum bending strain values of the ECC-SFP materials with three voids were 32.2%, 62.9% and 65.9% higher than for the SFP materials, respectively.

According to Figure 11c, the addition of ECC reduced the stiffness modulus. The stiffness modulus of the ECC-SFP materials was smaller than that of the SFP materials with the same void ratio of matrix asphalt mixture. Under the condition of 28 days of curing time, the stiffness modulus values of ECC-SFP were 18.9%, 24.4% and 28.9% smaller than for the SFP materials respectively.

In conclusion, adding ECC into cement mortar can increase the toughness of the SFP material, giving it a better deformation ability and resistance to cracking.

#### 3.2.3. Self-Healing Evaluation of ECC-SFP Materials

Two rutting specimens were molded for each void ratio of the matrix asphalt mixture, and the ECC mortar and ordinary mortar were grouted. After 7 days of curing, the rutting specimens were cut to obtain beam specimens. The beam specimens were divided into three groups for three curing conditions. The self-healing evaluation test was carried out according to the method in Section 2.4.4, and the test results are shown in Table 9.

It can be seen from Table 9 that curing conditions were the main factors affecting the self-healing properties of ECC-SFP materials. Consistent with the test results of ECC mortar, the recovery of specimens under three different curing conditions was different. The $HI_{\mathrm{mix}}$ of specimens cured in dry air were 4.4–7.3%. The $HI_{\mathrm{mix}}$ of specimens cured in moist air were 15.5–16.6%. The $HI_{\mathrm{mix}}$ of specimens cured in high temperature water were 24.8–26.2%. Secondly, as the void ratio of the matrix asphalt mixture increased, the $HI_{\mathrm{mix}}$ of specimens cured under dry conditions decreased, while the $HI_{\mathrm{mix}}$ of specimens cured under hot and humid conditions increased. Combined with the mix design results in Table 5, it can be inferred that the higher the amount of asphalt, the higher the $HI_{\mathrm{mix}}$ will be when cured in a dry environment. Asphalt is a major contributor to the self-healing properties of ECC-SFP materials in dry environments. 

The self-healing properties of ECC-SFP materials are related to ambient temperature and humidity. $HI_{\mathrm{mix}}$ increased by 8.2–12.2% at 20 °C when the ambient humidity increased from 10% to 100%. We can increase $HI_{\mathrm{mix}}$ by 9.3–9.6% under 100% humidity by raising the temperature from 20 °C to 60 °C. In hot and humid conditions, $HI_{\mathrm{mix}}$ can be increased by increasing the void ratio of the matrix asphalt mixture, but the effect is not obvious. Meanwhile, it can be seen from the deviation of the data that increasing the void ratio can increase the material uniformity. In conclusion, the void ratio of the matrix asphalt mixture should be 25–28%.

## 4. Conclusions

This paper studied the influence of Engineered Cementitious Composites (ECC) on the fluidity and strength of cement mortar, and studied the self-healing properties of ECC mortar and Semi-Flexible Pavement (SFP) materials grouted with ECC under three different curing conditions. According to the experimental results, the following conclusions can be drawn.

Firstly, the addition of ECC will reduce the fluidity of mortar. The fluidity of mortar decreases with the increase of ECC fiber dosage. To prepare ECC mortar which meets grouting requirements, a larger water–cement ratio should be used and the dosage of ECC should be controlled. In this study, cement mortar with a 0.23 water–cement ratio and less than 2% ECC fiber dosage was found to ensure fluidity.

Secondly, the addition of ECC gives the mortar better toughness. The flexural strength of ECC mortar is better than that of ordinary mortar, and the higher the ECC fiber dosage, the more significant the improvement of flexural strength.

Third, the grouted amount of ECC mortar affects the Marshall stability and flow value of the ECC-SFP material. As the void ratio of the matrix asphalt mixture increases, the amount of ECC mortar increases, and the Marshall stability and flow value increases.

Finally, curing conditions are the key factor affecting the self-healing property of ECC mortar and ECC-SFP materials. The materials have the best self-healing effect under a high temperature and humidity. When an ECC fiber dosage of 1% is used, $HI_{\mathrm{mor}}$ and $HI_{\mathrm{mix}}$ can reach 27.5% and 24.8%, respectively.

From the above conclusions, we can see that the use of ECC does give SFP materials certain self-healing properties, but the self-healing effect is still relatively poor. Improving the self-healing property of ECC-SFP materials should start with increasing the dosage of ECC, but this conflicts with the fluidity of the mortar. Therefore, ECC mortar is not suitable for using with the existing grouting methods. It is more important at this stage to optimize the grouting method and develop the ECC mortar grouting equipment for ECC-SFP materials. In terms of curing conditions, ECC-SFP materials should be used in areas with high temperatures and humidity. For example, the Guangdong-Hong Kong-Macao greater bay area in south China is a prime area for use of these materials. This bay area has a large amount of heavy traffic and perennial high temperatures and rain, which would maximize the usefulness of the self-healing properties of ECC-SFP materials. However, ECC-SFP materials are not suitable for cold and dry areas at middle and high latitudes.

In this study, some preliminary results regarding the self-healing properties of ECC-SFP materials were obtained; however, the fatigue durability at the macro-scale and the interface physicochemical properties of the materials at the micro-scale still need to be further studied.

## Figures and Tables

**Figure 1 materials-12-03488-f001:**
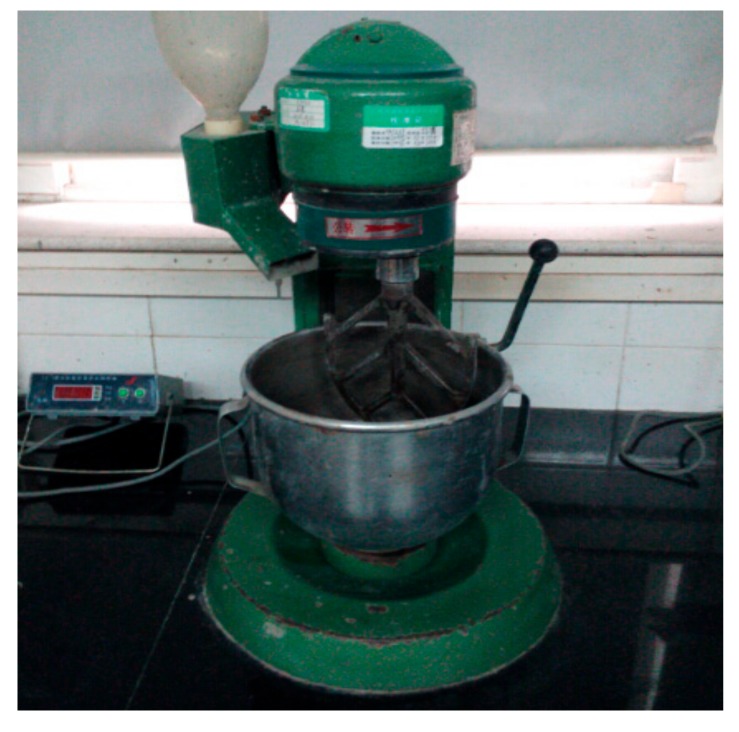
Cement mortar mixer.

**Figure 2 materials-12-03488-f002:**
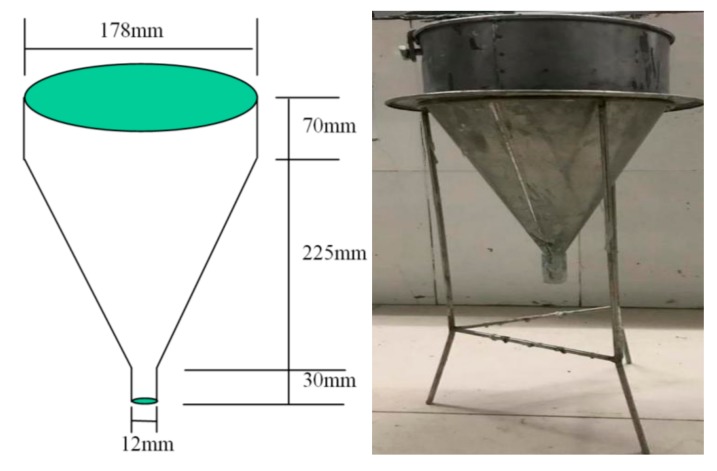
The funnel used for the test of fluidity.

**Figure 3 materials-12-03488-f003:**
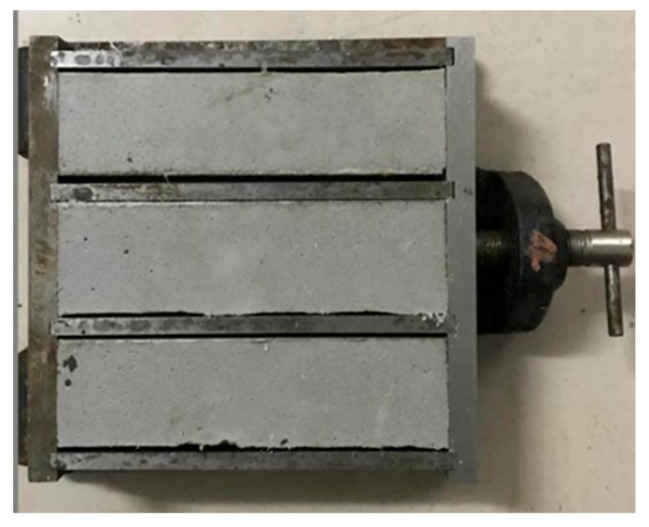
Cement mortar specimens.

**Figure 4 materials-12-03488-f004:**
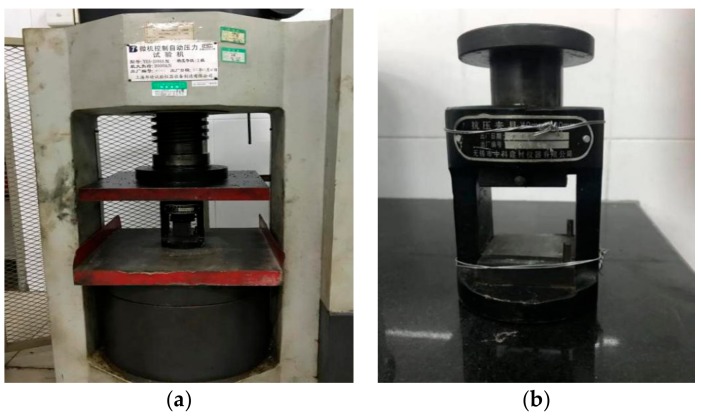
Settings of the compressive strength test. (**a**) Setup of compression test; (**b**) Test fixture.

**Figure 5 materials-12-03488-f005:**
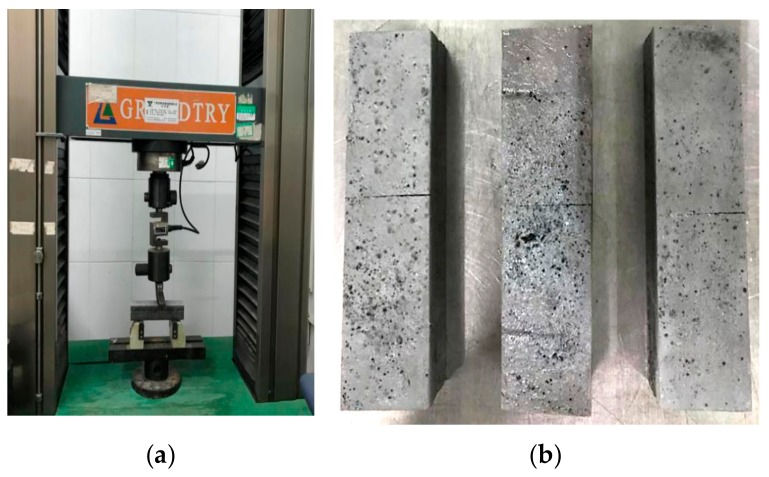
Preloading treatment of the specimens. (**a**) Preloading setup; (**b**) Preloaded specimens.

**Figure 6 materials-12-03488-f006:**
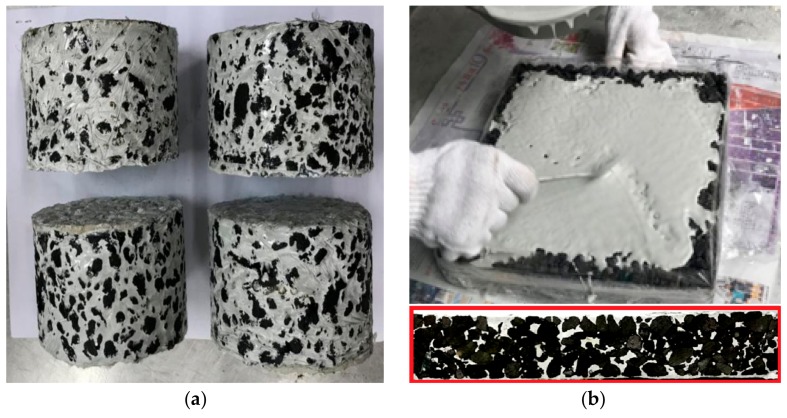
Grouting of the SFP material. (**a**) Marshall specimen grouting; (**b**) Rutting specimen grouting.

**Figure 7 materials-12-03488-f007:**
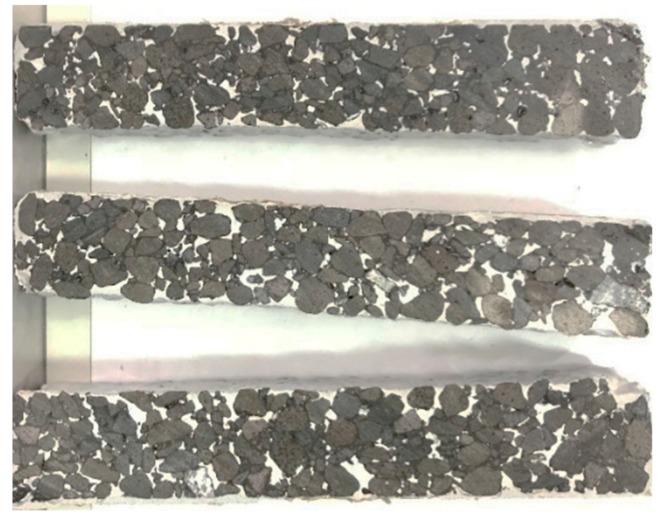
The beam specimens of the SFP materials.

**Figure 8 materials-12-03488-f008:**
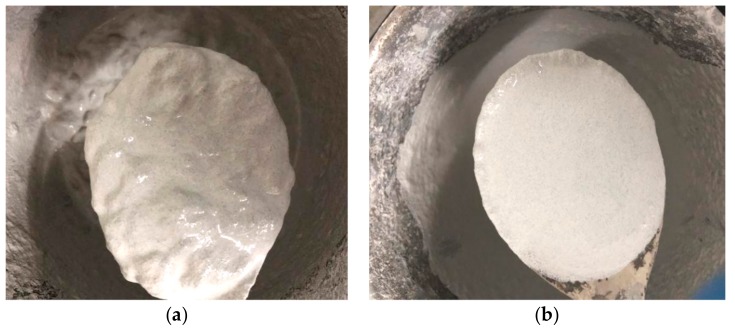
Flow state of ECC mortar. (**a**) Dosage 3%, w/c 0.24; (**b**) Dosage 2%, w/c 0.23.

**Figure 9 materials-12-03488-f009:**
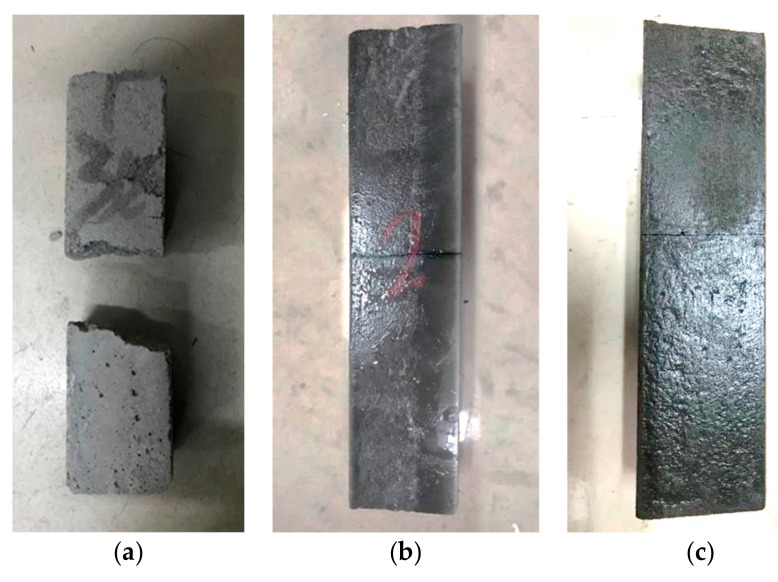
Failure modes of specimens with different ECC fiber dosages. (**a**) 0% Dosage; (**b**) 1% Dosage; (**c**) 2% Dosage.

**Figure 10 materials-12-03488-f010:**
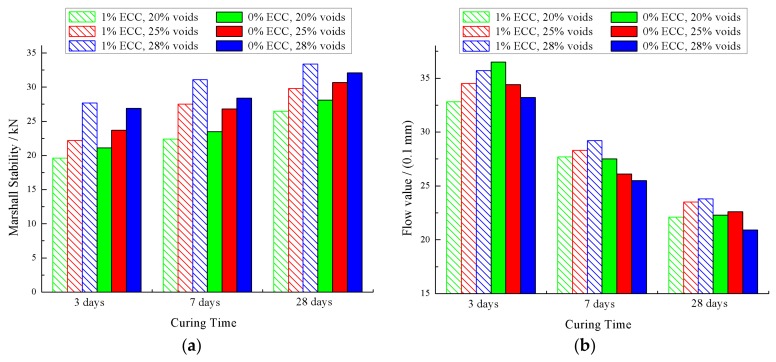
Marshall test results of SFP materials. (**a**) Results of the Marshall stability test; (**b**) Results of the flow value.

**Figure 11 materials-12-03488-f011:**
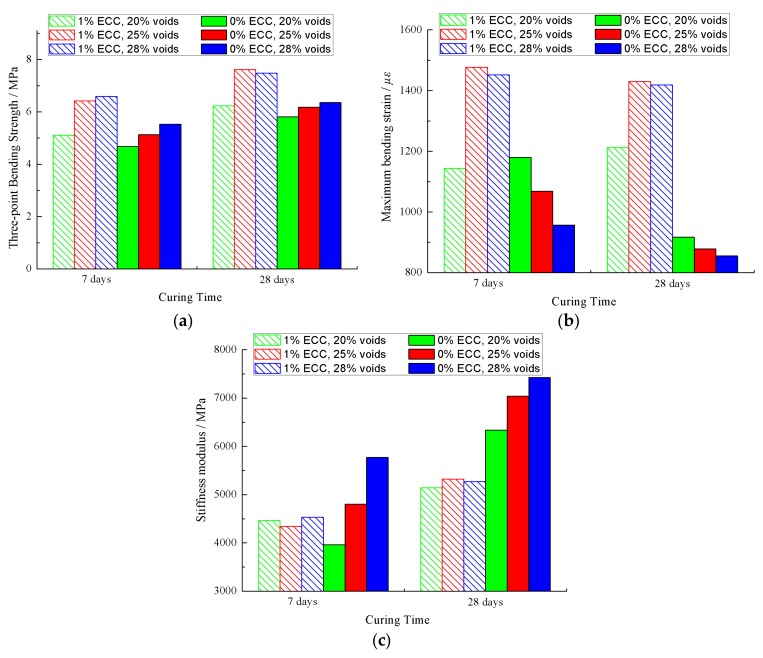
Results of the three-point bending test. (**a**) Results of three-point bending strength; (**b**) Results of maximum bending strain; (**c**) Results of stiffness modulus.

**Table 1 materials-12-03488-t001:** Technical indexes of 70# asphalt.

Technical Indexes	Unit	Test Results
Penetration at 25 °C	0.1 mm	70.2
Softening point	°C	49
Ductility at 10 °C	cm	51.3
Flash point	°C	335
Bitumen solubility (trichloroethylene)	%	99.8
Density at 15 °C	g/cm^3^	1.037

**Table 2 materials-12-03488-t002:** Technical indexes of the aggregates.

Aggregate Types	Technical Indexes	Unit	Test Results
Coarse aggregate	Crushing value	%	18
	Los Angeles abrasion	%	25
	Water absorption	%	0.17
	Apparent relative density	-	2.679
	Content of particles smaller than 0.075 mm	%	0.55
	Needle flake	%	12
Fine aggregate	Robustness	%	15
	Sand equivalent	%	68
	Apparent relative density	-	2.568

**Table 3 materials-12-03488-t003:** The components of the powder mortar.

Component	Cement	Sand	Fly ash	Rubber Powder	Water Reducer	Shrinkage Reducing Agent	Other Additives
Dosage %	38	25	25	3	0.5	0.7	7.8

**Table 4 materials-12-03488-t004:** Physical properties of the PVA fiber.

Density (g/cm^3^)	Elasticity Modulus (Gpa)	Tensile Strength (MPa)	Diameter (μm)	Elongation (%)	Length (mm)
1.30	30–50	1600–2500	15	6	5–10

**Table 5 materials-12-03488-t005:** Design of the matrix asphalt mixture.

Granular Composition	MA20	MA25	MA28
Voids (%)	20	25	28
Asphalt aggregate ratio (%)	4.0	3.6	3.3
16 mm	100	100	100
13.2 mm	94.6	94.6	94.6
9.5 mm	63.5	64.1	64.4
4.75 mm	14.7	12.4	11.9
2.36 mm	5.9	4.7	4.3
0.075 mm	3.5	3.0	3.0

**Table 6 materials-12-03488-t006:** Result of fluidity test of mortar.

Water-Cement Ratio	Dosage (%)	Fluidity (s)	Deviation
0.2	0	15.1	0.81
	1	16.3	0.65
	2	18.5	1.05
	3	21.6	1.56
0.21	0	13.6	0.41
	1	15.5	1.01
	2	17.3	0.81
	3	20.5	1.29
0.22	0	12.4	0.93
	1	14.8	1.25
	2	16.0	1.14
	3	19.6	1.26
0.23	0	10.8	0.53
	1	13.2	0.87
	2	15.1	0.85
	3	18.7	1.53
0.24	0	8.9	0.77
	1	11.2	0.71
	2	14.5	1.07
	3	17.5	1.24

**Table 7 materials-12-03488-t007:** Test results of the flexural strength and compressive strength of mortar.

Dosage/%	Curing Time/d	Flexural Strength (MPa)		Compressive Strength (MPa)	
		Mean	Deviation	Mean	Deviation
0	3	8	0.21	32	0.57
	7	10.6	0.18	52.3	0.55
	14	11.2	0.19	65.6	0.42
	28	13.3	0.14	75.9	0.68
1	3	8.3	0.25	29.8	0.44
	7	10.2	0.24	53.4	0.58
	14	12.8	0.64	65.9	0.75
	28	14.9	0.53	74.3	0.69
2	3	7.9	0.54	30.3	0.87
	7	10.3	0.49	51.7	0.74
	14	13.2	0.35	66.1	0.88
	28	15.5	0.47	73.6	0.61

**Table 8 materials-12-03488-t008:** Results of the mortar self-healing test.

Dosage (%)	Curing Condition	Specimen Code	${R_{f}}^{\prime }\;(\mathrm{MPa})$	$HI_{mor}$		
				Single Value (%)	Mean (%)	Deviation (%)
1	temp. 20 °C, humidity 10%	1-1	0.5	3.4	5.4	8.0
		1-2	1.1	7.4		
		1-3	0.8	5.4		
	temp. 20 °C, humidity 100%	1-4	2.1	14.1	13.7	5.5
		1-5	2.3	15.4		
		1-6	1.8	12.1		
	temp. 60 °C, in water	1-7	4.2	28.2	27.5	6.2
		1-8	4.3	28.8		
		1-9	3.8	25.5		
2	temp. 20 °C, humidity 10%	2-1	1.1	7.1	6.7	3.4
		2-2	0.8	5.2		
		2-3	1.2	7.7		
	temp. 20 °C, humidity 100%	2-4	2.8	18.1	16.1	7.6
		2-5	2.5	16.1		
		2-6	2.2	14.2		
	temp. 60 °C, in water	2-7	4.4	28.4	30.3	5.8
		2-8	4.9	31.6		
		2-9	4.8	31.0		

**Table 9 materials-12-03488-t009:** Self-healing test results of the ECC-SFP materials.

AC Type	Curing Condition	Specimen Code	${R_{B}}^{\prime }\;(\mathrm{MPa})$	$HI_{mix}$		
				Single Value (%)	Mean (%)	Deviation (%)
MA20	temp. 20 °C, humidity 10%	1-1	0.46	7.4	7.3	2.5
		1-2	0.52	8.3		
		1-3	0.38	6.1		
	temp. 20 °C, humidity 100%	1-4	1.08	17.3	15.5	5.9
		1-5	0.87	14.0		
		1-6	0.94	15.1		
	temp. 60 °C, in water	1-7	1.53	24.6	24.8	8.8
		1-8	1.42	22.8		
		1-9	1.68	27.0		
MA25	temp. 20 °C, humidity 10%	2-1	0.53	7.0	5.7	2.6
		2-2	0.36	4.7		
		2-3	0.42	5.5		
	temp. 20 °C, humidity 100%	2-4	1.25	16.4	16.1	2.1
		2-5	1.14	15.0		
		2-6	1.29	17.0		
	temp. 60 °C, in water	2-7	1.86	24.4	25.9	6.8
		2-8	2.13	28.0		
		2-9	1.93	25.4		
MA28	temp. 20 °C, humidity 10%	3-1	0.38	5.1	4.4	0.9
		3-2	0.28	3.7		
		3-3	0.33	4.4		
	temp. 20 °C, humidity 100 %	3-4	1.28	17.1	16.6	4.3
		3-5	1.12	15.0		
		3-6	1.33	17.8		
	temp. 60 °C, in water	3-7	2.02	27.0	26.2	5.0
		3-8	1.82	24.3		
		3-9	2.03	27.1		

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
