# Peer review of "Study of the Self-Healing Performance of Semi-Flexible Pavement Materials Grouted with Engineered Cementitious Composites Mortar based on a Non-Standard Test"

_materials, 2019, doi:10.3390/ma12213488_

Round 1
Reviewer 1 Report
This paper tries to study the self-healing behavior of ECC and ECC-SFP mixtures. It could be an interesting topic as the healing of ECC-SFP interface crack, if there is any, has not been well studied. But this paper just compares the mechanical properties of healed ECC and healed ECC-SFP, and did not go deep into the mechanisms. It does not demonstrate the sealing of the cracks, while the mechanical recovery may result from many other reasons: continued hydration of the matrix, change of asphalt under elevated temperature...
The introduction just listed the literature on SFP, not really trying to systematically identify the knowledge gaps, especially those in self-healing.
Some expressions in this paper is confusing, such as "ECC dosage" (I believe the authors are trying to say "fiber dosage".
Author Response
Thank you very much for your comments on this paper! Your suggestion points out the inadequacy of our research. At present, the research we have carried out is only the regular phenomenological results, and has not yet reached the microscopic and microscopic level. The study on self-healing mechanism of materials is what we need to carry out in the future. At present, we believe that the self-healing mechanism of ECC-SFP materials should be developed from two aspects:
The influence of ECC on crack resistance and crack width control. The effect of cement secondary hydration on crack repair. Study on the contribution of asphalt and cement mortar to self-healing. We think this study is very challenging.
According to your suggestion, we added references and changed "ECC dosage" to "ECC fiber dosage".

Reviewer 2 Report
The paper was passed on the condition of minor correction.
line 23: order --> order to
line 50: Koting[11] ?
Author Response
Thank you very much for your comments on this paper! According to your revision suggestions, we have re-checked the grammar of the article and the corresponding relationship of references.

Reviewer 3 Report
Good paper!
Author Response
Thank you very much for your approval of this study! Your appreciation inspires us! We will continue to work hard in this area to continue in-depth research!

Reviewer 4 Report
This manuscript presents the results of an interesting study about the effect of an additive and different curing conditions on the mechanical properties of semi-flexible pavement materials. However, if fails in convincing me about the concept of self-healing used by the authors. The testing protocol is not supported in the literature (or in standardized procedures) and the may not accurately measure the healing effect of a rest period. The ratio between the strength of preloaded specimens and the strength of specimens not subjected to any preload does not necessarily represent the self-healing properties of that material. Thus, I would recommend the authors to revise the manuscript, changing its title and scope and resubmitting it again for a new revision process.
Apart from this very important issue that prevents me from accepting its publication, the manuscript is well written and only a few typographical mistakes were found, which should be corrected in the next submission.
Author Response
Thank you very much for your comments on this paper! Indeed, most current definitions of self-healing are based on the percentage of a material's properties that can be recovered after damage. We used the ratio between the strength of preloaded specimens and the strength of specimens not subjected to any preload to evaluate the self-healing effect of the material. This practice is taking into account that the strength of cement mortar will increase with the increase of curing time. If only preload specimens are used for evaluation according to the practice of asphalt mixture, the following problem will occur:
The strength of cement mortar varies from 30-50% between the two tests in 7d and 28d. And the calculated self-healing ratio includes the contribution of cement hydration.
Therefore, we believe that the strength of non-preload specimens at 28d should be compared to better evaluate the properties of materials. Based on your comments, we also realize that this is not a routine self-healing trial. But the experiment also involved self-healing, so we changed the title of the article:
Study of the Self-Healing Performance of Semi-Flexible Pavement Materials Grouted with Engineered Cementitious Composites Mortar based on a Non-standard Test

Reviewer 5 Report
As a reviewer, I would like to thank the authors for their effort and for the chosen topic. However, I suggest addressing the following minor comments to improve it. Noticed that I decided to separate comments by the main sections of the manuscript they belongs to and not every section need to be changed.
Abstract
The reading of this part is comfortable to read and it is well written and I think every important thing of the research is mentioned so congratulations.
Introduction
In the introduction I miss a paragraph mentioning the self-healing aided methods as magnetic induction or microwaves. Doing a quick research I found this study which is also from this journal:
“Vila-Cortavitarte, M.; Jato-Espino, D.; Castro-Fresno, D.; Calzada-Pérez, M.Á. Self-Healing Capacity of Asphalt Mixtures Including By-Products Both as Aggregates and Heating Inductors. Materials 2018, 11, 800.”
I strongly recommend you to add this reference to your manuscript.
Experimental Design
As well as is some other parts I would suggest you to avoid section titles with no text within them as in lines 101-103
In tables, I suggest you to indicate the unit in brackets and not after a bar “/” in order not to see it as a divisor. In this section it happens in table 4 and 5.
Avoid in this section and along the manuscript abbreviations in the section titles as ECC or SPF, even I would recommend you to fully write the meaning of each abbreviation the first time it is written in each section.
If would recommend you to add a brief description of the statistical methodology used to get the deviations on next section tables.
Results and Discussion
As at the beginning of previous section, add some text between section and subsection titles (lines 237-239) in order not to start in such an abrupt way.
If possible, avoid breaking tables in two pages because it makes the comprehension more difficult. Nevertheless, this will be probably accomplished by the Journal Final Edition if the manuscript is finally accepted.
Again, avoid abbreviation in titles.
In line 302, SEP is written instead of SFP.
Both in table 8 and table 9 “Devotion” is written instead of “Deviation“
Again, in tables 6,7,8 and 9 I suggest you to put units in brackets
Conclusions
Although they have been written several times in the manuscript and may seems redundant, rewrite the full names of the abbreviation used in the conclusion the first time you used thinking on the potential readers who may reach your article and go straight into the conclusions section.
Author Response
Thank you very much for your comments on this paper! According to your valuable advice, our replies are as follows:
Thank you very much for your approval of the abstract! Your appreciation strengthens our confidence to continue our research! We have added your recommended paper to the references. Thanks for your advice! We have added the titles in lines 101-103. According to your suggestion, we replaced the "/" in table 4 and 5 and re-checked the other tables. The full names of ECC and SPF are given in the summary and introductory sections and later abbreviated. We also checked the title and changed the abbreviation of the title. Thank you for your suggestions, which are very helpful to improve our paper writing. Because the statistical method used in this study is simple to calculate the mean and deviation, we think that no further explanation is needed. If other methods have been used, we will consider clarification. According to your advice, we add some text between the title in order not to start in such an abrupt way. We will pay more attention to the layout of the table to make it easier for reviewers to read. We have modified the mistake in line 302, table 8 and table 9. Thank you for your care and patience.9. According to your suggestion, we write the full name of the abbreviation used in the conclusion.

Round 2
Reviewer 4 Report
The authors made an effort to improve the quality of the manuscript and changed the title to reflect its content better. However, they should have included a better explanation of the reason why they used the ratio between the strength of preloaded specimens and the strength of specimens not subjected to any preload to measure the healing effect of a rest period. I understood the explanation they included in response to my previous comments, but they did not include such an explanation in the manuscript.
Thus, I would recommend the publication of the manuscript after the authors include the explanation in the paper.
Author Response
Thanks for the attention and patience of the reviewer! In accordance with the expert's comments and suggestions, we have supplemented the explanation for the use of non-standard tests. This make us more understand the definition of self-healing. It also reminds us that we should consider the problem from a more comprehensive perspective when designing experiments.
